# Composition and Function of Telomerase—A Polymerase Associated with the Origin of Eukaryotes

**DOI:** 10.3390/biom10101425

**Published:** 2020-10-08

**Authors:** Petra Procházková Schrumpfová, Jiří Fajkus

**Affiliations:** 1Laboratory of Functional Genomics and Proteomics, National Centre for Biomolecular Research, Faculty of Science, Masaryk University, Kotlářská 2, CZ-61137 Brno, Czech Republic; fajkus@sci.muni.cz; 2Mendel Centre for Plant Genomics and Proteomics, Central European Institute of Technology, Masaryk University, Kamenice 5, CZ-62500 Brno, Czech Republic; 3The Czech Academy of Sciences, Institute of Biophysics, Královopolská 135, 612 65 Brno, Czech Republic

**Keywords:** telomerase, evolution, telomerase RNA (TR), telomerase reverse transcriptase (TERT), plant TERT, plant TR

## Abstract

The canonical DNA polymerases involved in the replication of the genome are unable to fully replicate the physical ends of linear chromosomes, called telomeres. Chromosomal termini thus become shortened in each cell cycle. The maintenance of telomeres requires telomerase—a specific RNA-dependent DNA polymerase enzyme complex that carries its own RNA template and adds telomeric repeats to the ends of chromosomes using a reverse transcription mechanism. Both core subunits of telomerase—its catalytic telomerase reverse transcriptase (TERT) subunit and telomerase RNA (TR) component—were identified in quick succession in Tetrahymena more than 30 years ago. Since then, both telomerase subunits have been described in various organisms including yeasts, mammals, birds, reptiles and fish. Despite the fact that telomerase activity in plants was described 25 years ago and the TERT subunit four years later, a genuine plant TR has only recently been identified by our group. In this review, we focus on the structure, composition and function of telomerases. In addition, we discuss the origin and phylogenetic divergence of this unique RNA-dependent DNA polymerase as a witness of early eukaryotic evolution. Specifically, we discuss the latest information regarding the recently discovered TR component in plants, its conservation and its structural features.

## 1. Telomerase Activity

Telomerase reverse transcriptase is a specific nucleoprotein enzyme complex that solves the problem that conventional DNA replication machinery can not fill in the gap after the removal of the RNA primer of a most distal Okazaki fragment at the 5’-terminus of the lagging strand. This results in a loss of a small portion of chromosomal DNA. This phenomenon is called the end-replication problem (Figure 1), first defined by Olovnikov [1]. Moreover, the ends of eukaryotic chromosomes—telomeres—must be long enough to assemble a protective nucleoprotein “capping” structure that can distinguish a natural terminus from an unrepaired chromosomal break. Dysfunctional telomeres may trigger genome instability, cell cycle arrest, and—at least in humans—replicative cell senescence and apoptosis (reviewed in [2,3]).

In humans, telomerase activity is detected in all early developmental stages and increases progressively with advancing embryonic stages. After the completion of organogenesis in the human fetus, telomerase is expressed only in proliferating tissue-specific stem cells (e.g., bone marrow progenitor cells and neural stem cells), while telomerase activity in somatic cells is downregulated (reviewed in [7]). However, a tendency to repress telomerase in mammalian somatic tissues was described only for mammalian species of weight greater than 1 kg; e.g., laboratory mice have a constitutive telomerase. It was proposed that in long-lived species, telomerase downregulation may have evolved to limit cell proliferation and reduce the risk of cancer. Correspondingly, ca. 90% of all human tumors display telomerase reactivation to achieve cellular immortality [8].

Telomerase, as a primary mechanism for telomere maintenance, is also conserved in plants. Analogous to mammals, telomerase activity is suppressed in terminally differentiated tissues, e.g., mature leaves or stems. Active telomerase is detected in organs and tissues such as seedlings, shoot and root tips, young and middle-age leaves, flowers, and floral buds with proliferating meristematic cells [7,9,10,11,12,13].

Telomerase carries, in addition to its protein catalytic subunit (telomerase reverse transcriptase; TERT), its own RNA templating subunit (telomerase RNA; TR) (reviewed in [14]). The expression of human TERT is strictly controlled at the transcript level and closely associated with telomerase activity, which suggests that TERT in humans is the primary determinant of enzyme activity [15]. In most human tissues, TR is ubiquitously expressed regardless of telomerase activity, and therefore, it has been considered by some authors as a non-limiting factor for telomerase activity [16]. However, telomerase activity in human T lymphocytes has been reported to relate to TR levels but not TERT protein levels [17,18].

In plants, contrary to most human cells, the expression of the TR subunit, recently characterized by Fajkus et al. (2019) [19], follows a tissue-specific pattern similar to that which is typical for the expression of TERT. In thale cress (*Arabidopsis thaliana*), the highest *TERT* mRNA levels were detected in flower buds, lower transcript levels were detected in seedlings and young leaves, and the lowest levels were observed in aged leaves [11]. Similarly, *TR* transcripts were most abundant in flower buds and 7-day-old seedlings. Markedly lower yet detectable levels were observed using RT-qPCR in 3-week-old seedlings and young leaves. Absolute levels of *TR* transcripts were 60–70 times higher than *TERT* mRNA levels [19]. Whole-mount in situ hybridization detected *TR* transcripts in the primary root and lateral root apices of 3-week-old seedlings and in cultured cells, but in other tissue samples, using northern hybridization, no TR signal was found [20]. Levels of *TR* or *TERT* transcripts correlate strongly with the telomerase activities observed in various plant tissues [7,11].

## 2. The Origin of Telomerase

The telomerase RNA-dependent DNA polymerase arose specifically within the eukaryotic lineage and was able to successfully solve the end-replication problem of linear chromosomes that leads to telomere shortening [21]. Telomeres are composed of short non-coding tandem repeat units, the length of which can significantly vary among diverse taxons. The lengths of telomere arrays can also vary at the level of the species or ecotypes (reviewed in [22,23]). The human-type (TTAGGG)_n_ telomeric sequence is conserved across several eukaryotic ‘supergroups’ [24], including the Amorphea supergroup with metazoan and fungal species (reviewed in [25,26]). At the same time, several exceptions are known across the Amorphea supergroup, e.g., in insects (TTAGG)_n_ [27,28], in Nematodes (TTAGGC)_n_ [29] or in several fungal genera, where very complex or irregular telomeric motifs were described [26,30] (Figure 2). Additionally, the telomeres of some insects are constituted with unusual telomeric motifs, or even with telomeric repeats that consist of arrays of non-long-terminal-repeat (non-LTR) retrotransposons [31,32,33]. The *TERT* gene disappeared from the genomes of some insects with non-LTR retrotransposons, as in the vinegar fly (e.g., *Drosophila melanogaster*). In silkworms (*Bombyx mori)*, *TERT* is very weakly expressed in various tissues, telomerase activity is barely detectable, and retrotransposition is required to maintain the length of chromosome ends [33,34,35,36].

In the Archaeplastida supergroup that includes the land plants, mosses, red algae and green algae, the telomere is mostly composed of (TTTAGGG)_n_ repeats, first described in *A. thaliana* Arabidopsis-type repeats [40]. Despite this, the human-type telomere repeat is shared by several plant taxa from the order Asparagales [41], including species of the Allioideae subfamily, except for the Allium genus [42], where a more complex telomeric sequence (CTCGGTTATGGG)_n_ was described [43]. An unusual telomeric motif (TTTTTTAGGG)_n_ was found in the genus *Cestrum* (Solanaceae) [44], and in some species from the carnivorous genus *Genlisea* (TTCAGG and TTTCAGG) [45]. Outside of land plants, repeats other than the Arabidopsis-type were characterized in some algae and glaucophyte species ((AATGGGGGG)_n_, (TTTTAGGG)_n_, (TTTTAGG)_n_ etc.) [22,38,46]. Unlike insects, even the unusual plant telomeric sequences characterized so far are synthesized by telomerases using TRs with corresponding template regions [19,43,44,47].

Before the linear chromosomes of eukaryotes emerged, ∼1 Gy ago, circular chromosomes had been successfully used for 2 Gy in eubacteria and archaea, and they still predominate in most bacterial forms [48]. In a concept elaborated by E. V. Koonin [37], the origin of linear chromosomes, telomeres and telomerase is associated with the invasion of archaeal hosts by an alpha-proteobacterial progenitor that resulted in mitochondrial endosymbiosis and the invasion of Group II self-splicing introns, the most ancient genetic entities (Figure 2). Group II introns were suggested as eukaryotic evolutionary ancestors of retrotransposons and spliceosomal introns. They consist of a catalytically active intron RNA and an intron-encoded reverse transcriptase (RT), which is related to non-LTR-retrotransposon RTs and assists splicing by stabilizing the catalytically active RNA structure (reviewed in [49]). The invasion of Group II introns resulted in the evolution of spliceosomal introns, the compartmentalization of the majority of genetic information in the nucleus and the linearization of chromosomes. At the same time, solution of the end-replication problem was hinted at by a mechanism pre-existing in a primordial pool of ‘virus-like’ genetic elements in the earliest stages of life’s evolution [37]. It seems that incipient eukaryotes, due to the presence of Group II sequences, could have had stable linear chromosomes without the need for telomerase- or telomere-specific proteins [48].

The TERT component of telomerase is highly conserved, having a centrally positioned reverse transcriptase motif (RT domain) [50,51,52]. The presence of TERT was detected in early branching eukaryotes [53,54]. It was proposed that telomerase originated as an ancient reverse transcriptase (RT) that internalized a primitive template-bearing RNA during early eukaryotic evolution, and later evolved into modern telomerase ribonucleoproteins (RNPs) with various indispensable and stably TR-associated components [55,56]. In accordance with this notion, the TERT subunit has the ability to bind not exclusively the RNA molecule that provides the template sequence for DNA synthesis (TR), but also various non-telomeric sequences [57,58]. The conserved RT motifs between TERT and other RTs indicates that the TERT protein is closely related to RTs from the group of Penelope-like Elements (PLEs) and non-long-terminal-repeat (non-LTR) retrotransposons [55]. PLEs have an RT that lacks endonuclease activity, and it is plausible that ancient retrotransposons similar to these terminal PLEs might be the progenitors of TERT proteins [56].

Interestingly, besides the telomerase-dependent mechanism of telomere elongation, yeast, mammalian and plant cells can use alternative mechanisms of lengthening of telomeres (ALT) based on homologous recombination (HR). ALT usually results in telomeres that are highly heterogeneous in length and sequence [59,60,61]. In plants, the ALT mechanism is activated in mutants with telomerase dysfunction, and possibly also during the earliest stages of normal plant development [59,60,61,62].

It is suggested that the canonical telomeric repeats have been changed or lost independently several times during evolution [36,46], and telomerase may have even occasionally been lost [36]. It would be interesting to further investigate the occurrence of unusual telomeric motifs, the co-evolution of genes encoding core telomerase subunits (TERT and TR), and the replacement of telomerase by telomerase-independent ALT systems (probably the ancestral telomere maintenance tools) in cases where there is evolutionary loss of telomerase.

## 3. RNA Subunit of Telomerase

The non-coding RNA serving as telomerase RNA (TR), also known as TER or TERC, contains the template region for the addition of telomeric repeats. TR is highly divergent, compared to TERT, ranging in size from ∼150 nucleotides (nt) in ciliates to over 3000 nt in yeasts [63]. TR has a number of conserved structural domains, consisting of the template/pseudoknot (t/PK) domain, template boundary element (TBE) and stem-loop region. In humans, the stem-loop region contains conserved structural domains: a conserved region 4/5 (CR4/5); the 3′ H/ACA (H-box (consensus ANANNA); an ACA-box (ACA) domain; and a Cajal body box (CAB-box) motif (reviewed in [64,65]) (Figure 3a). Although the gene coding telomerase RNA had already been identified in *Tetrahymena* in 1989, and in humans in 1995 [14,66], it took another three decades for the first bona fide plant *TR* genes to be characterized, in 2019 [19]. Interestingly, the unusual length of the Allium telomere repeat unit (12 nt) [43] was used to identify candidate TRs, not only in *Allium* species but subsequently across the phylogeny of land plants with either canonical or unusual types of telomere repeats [19]. Previously characterized TER1 and TER2 in *A. thaliana* were shown not to act as telomerase RNAs [19,67], and the original paper describing them has been retracted [68,69].

In mammals, telomerase RNA belongs to the family of small nucleolar (snoRNAs) and small Cajal body (scaRNAs) RNAs [76,77]. Both snoRNAs and scaRNAs are encoded in introns and transcribed by RNA polymerase II (RNA Pol II), along with their host structural genes [78,79,80]. The human *TR* primary transcript is synthesized by RNA Pol II, capped on its 5′ end with a monomethylguanosine (MMG) cap that is further methylated to N2, 2, 7 trimethylguanosine (TMG) cap [81,82,83], internally modified, and processed at its 3′ end to generate the mature, functional TR (reviewed in [65,84]). Several structural motifs and the formation of the overall tertiary structure of TR are needed for a proper interaction with the TERT subunit (reviewed in [85]). Although TERT can bind to TR through the t/PK domain alone, additional binding with the CR4/5 domain is required [86,87].

Like many other polymerases, telomerase catalyzes nucleotide addition to the 3′ hydroxyl group of a primer, forming a product–template duplex. Accurate telomeric repeat synthesis depends on the strict boundaries of a template region within TR, which functions as a STOP signal in the telomerase extension step (reviewed in [70]). In humans, to synthesize 6 nt telomeric repeats of the human-type telomeric motif, telomerase anneals five nucleotides of its 11 nt long template region with terminal nucleotides of the telomere DNA, and extends it with 6 nt complementary to the rest of the template region [88].

Telomerase processivity requires repeated cycles of annealing, synthesis, translocation and re-annealing of the substrate DNA–TR base-pairing [70]. Telomerase remains associated with substrate DNA even when DNA–RNA base-pairing is disrupted, however the exact mechanism was unknown [89,90,91]. Recently, the details of processive telomerase catalysis were revealed using high-resolution optical tweezers. The authors demonstrated that a stable substrate DNA binding at an anchor site within telomerase facilitates the processive synthesis of telomeric repeats, which results in synthesizing multiple telomeric repeats before releasing them in a single step. The product DNA synthesized by telomerase can be recaptured by the anchor site or folded into G-quadruplex structures [92].

It remains controversial whether the active telomerase enzyme in humans functions as a dimer (TR and TERT) or only as a monomer of each subunit [93,94,95,96]. In contrast to the human telomerase complex, the affinity-purified telomerase from *Tetrahymena* is monomeric [97]. The possible dimerization of telomerase was also suggested in plants. Dimerization, modulated by a conserved TRBD domain from *A. thaliana* TERT that is able to interact separately with the N-terminal fragments and itself, was observed using yeast two-hybrid analysis of interactions [98].

While the mechanism of the telomerase catalytic cycle may be similar among telomerases from different kingdoms, the recent characterization of plant *TRs* across the whole land plant phylogeny revealed some features distinct from TRs described in mammals or fungi. First, plant *TRs* are transcribed with RNA polymerase III (RNA Pol III) [19], similarly to *TRs* in ciliates, while mammalian or yeast TRs are RNA Pol II products [55]. The closer relationship between TR biogenesis in ciliates (supergroup TSAR) and plants (supergroup Archaeplastida) on the one hand, and fungi and animals (supergroup Amorphea) on the other hand, corresponds to current versions of the phylogenetic tree of eukaryotes (which is supported by phylogenomic studies), and respective ancestral supergroups [24]. Plant *TRs* show relatively conserved structures in their RNA Pol III promoters (so-called Type 3 RNA Pol III promoter [99] with a typical Upstream Sequence Element (USE) and TATA box). Further, *TRs* in land plants have a monophyletic origin [19]. This contradicts the previous paradigm, according to which relatively conserved TERT subunits associate with very diverse—and unrelated—RNAs [57,58].

Interestingly, the template regions of plant TRs are of relatively diverse lengths. They are mostly of the length corresponding to one and one half of the telomere repeat, which allows for substrate DNA annealing. The template regions may, however, also be shorter (e.g., in *A. thaliana* TR, whose telomere repeat unit is 7 nt and template region is only 9 nt long) or longer (e.g., wild carrots (*Daucus carota*) TR harbor template region of 14 nt, corresponding to two complete telomere repeat units of 7 nt) [19]. It is important to note that the authentic, functional part of the template region may be shorter than the putative predicted template (the region complementary to the synthesized telomere repeat), as it is delimited by secondary structure elements in TRs. Some other secondary structure motifs of TRs—e.g., the pseudoknot structure downstream of the template region—seem conserved among animal, plant and fungal TRs [55,64,71].

Whether a primary transcript of plant telomerase RNA is generated in a way similar to that in ciliates (synthesized by RNA Pol III, not spliced, leaving a 3′ polyuridine tail [64,100]), and which motifs, domains or stems of TR are involved in TR–TERT interactions, needs to be clarified. Moreover, the functions of TR expand far beyond its templating function, as it forms a flexible scaffold that functions in correct telomerase RNP assembly.

## 4. TERT Subunit of Telomerase

Most of the catalytic subunits of telomerase, TERTs, including the human and plant TERTs, can be classified into three major parts. At the N-terminus are positioned telomerase-specific motifs (N-terminal domain), reverse transcriptase motifs (RT domain) are positioned centrally, and at the C-terminus of the TERT protein are localized conserved motifs—these are more or less specific for particular groups of organisms (C-terminal extension, CTE) (Figure 3b).

The central RT domain is the catalytical part of the enzyme, and this contains seven evolutionarily conserved RT motifs (1, 2, A, B′, C, D and E motifs). This domain is organized into two subdomains—the “fingers” involved in nucleotide binding and processivity, and the “palm” providing the polymerase catalytic residues and DNA primer grip [50,51,94,101].

There are two main domains recognized within the N-terminal part: the telomerase essential N-terminal domain (TEN, also known as the RNA interaction domain 1 (RID1)) [102,103] and the RNA-binding domain (TRBD) (reviewed in [75]). Moreover, a variable linker physically and functionally separates these two domains and has been shown to be biologically essential for the function of TERT (reviewed in [103]). The nucleus localization-like signal (NLS), placed between the TEN and TRBD domains, is responsible for the nuclear import of TERTs [104].

The TEN domain has both DNA-binding and nonspecific RNA-binding properties, and may also stabilize short RNA–DNA duplexes during telomere extension, i.e., repeated cycles of telomerase annealing, synthesis, translocation and re-annealing.

Despite poor sequence homology, the CTE part is almost universally conserved, although several roundworm species appear to lack this structure entirely [102]. The crystal structure of the human CTE domain identified three highly conserved regions within the CTE region [105]. It is proposed that CTE is involved in promoting telomerase processivity, in regulating telomerase localization, and is involved in the differential binding of DNA, but not in essential catalytic functions, as reviewed in [106].

It was proposed that TERT in metazoan ancestors possesses all three major parts and 11 canonical motifs—GQ, CP, QFP, T motifs within the N-terminal part, and 1, 2, A, B′, C, D and E motifs within the RT part. However, GQ and CP motifs might be missing in some beetles (e.g., *Tribolium castaneum*) [103] or unicellular relatives of metazoans (e.g., Trypanosoma sp.). Similarly, the plant TERTs possess three major parts, 11 canonical motifs and a CTE part [75].

Although the TERT protein is highly conserved, the gene structure differs among the group of organisms, as they differ in exon/intron organization. In ciliates, species with 1 exon (e.g., *Euplotes aediculatus*) to 19 exons (*Tetrahymena thermophila*) were identified [75]. Contrary to ciliates, the *TERT* exon–intron structure is conserved across the Vertebrata. Mammalian *TERT* has 16 exons, whereas *TERTs* in non-mammalian vertebrates have anywhere between 14 and 17 exons [50,103,107,108,109]. Among plants, *TERT* genes with 12 exons are highly conserved [75]. Although most eukaryotes, including humans, harbor a single *TERT* gene, in polyploid plant species, as in allotetraploid *Nicotiana tabacum* (tobacco), multiple *TERT* paralogs exist that are differentially regulated [10,110].

## 5. Telomerase Regulation

As described above, the primary determinant for telomerase enzyme activity in humans seems to be a strictly controlled transcription level of the TERT subunit [15], rather than the expression of the TR subunit, which is ubiquitously expressed in almost all tissues regardless of telomerase activity [16].

Many studies have dissected the mammalian *TERT* promoter and identified cis-elements (E-boxes, GC motifs, ETS domain) bound by general transcription factors (TFs), such as c-MYC, NF-κB, STAT3, SP1 or ETS2 (Figure 4a) (reviewed in [101,111,112,113]). Moreover, these general TFs can be regulated by a number of other proteins; e.g., human RuvBL2 (reptin) regulates the c-MYC-dependent transcription of *TERT* [114]. The core functional promoter essential for the transcriptional activation of human *TERT* in cancer cells was suggested to be in the 181-bp [115] or 208-bp fragment [116], respectively, upstream of the transcription start site. In plants, the 336 bp long promoter region of the *TERT* promoter seems to be essential for the successful complementation of telomerase and reversion of the short telomere phenotype in *tert −/− A. thaliana* plants (Table 1a) [13,117]. Crhák et al. showed the efficient and tissue-specific control of telomerase reconstitution. At the same time, the results have shown that the level of Arabidopsis *TERT* transcript is not the sole determinant for the successful restoration of the telomeric function of telomerase, which suggests posttranscriptional control of telomerase expression [117]. Moreover, the restoration of telomerase activity, as evaluated in complemented plant extracts in vitro, did not always correlate with the ability to restore telomere maintenance in planta.

The wild-type promoter of the human *TERT* gene is often silenced by the repressive trimethylation of Lys27 in histone H3 (H3K27me3) modification. Consistent with this finding, the mutated human *TERT* allele is marked by the active histone marks H3K4me2, H3K4me3 and acetylated histone H3 Lys9 (H3K9ac) [118,119]. Analysis of the epigenetic states of the *TERT* gene in Arabidopsis telomerase-positive and telomerase-negative tissues revealed differential levels of H3K27me3, the mark of developmentally silenced heterochromatin regions in plants, whereas euchromatin-specific marks (H3K4me3 and H3K9Ac) were approximately at the same levels in all tissues [11]. The striking stability of the epigenetic status of the *TERT* promoter in Arabidopsis may reflect a unique attribute of plants—their totipotency, which is in accordance with the reversible and dynamic character of telomerase silencing [120].

Additionally, *TERT* transcripts in many animal species, including vertebrates, insects and nematodes, are alternatively spliced [75,103,146,147]. Specific patterns of *TERT* mRNA variants expressed in humans and rodents during development indicate that splicing events are not random and could have a physiological function (Figure 4a) [75,148,149,150,151]. Human *TERT* pre-mRNAs in early development can be spliced into 22 isoforms, while telomerase activity is associated only with the full-length *TERT* mRNA (reviewed in [152]). Some of the alternate human *TERT* mRNA forms (e.g., the minus alpha-variant) may not only be catalytically inactive, but even show a dominant negative inhibition of telomerase activity [153]. Correspondingly, differential patterns of *TERT* mRNA splicing were observed between normal (fetal human colon, FHC) and adenocarcinoma colon (HT-29) cells during their sodium butyrate-induced differentiation. The higher abundance of the minus alpha-variant of *TERT* mRNA was observed in FHC cells, which may be involved in the more rapid loss of telomerase activity in these cells during differentiation [154]. Spliced variants may also have non-canonical roles, for example in cell proliferation [125,155]. Alternative splicing was also described in many plant species (e.g., *A. thaliana*, *Oryza sativa* (rice), *Iris tectorum* (roof iris)) (reviewed in [75]). Short isoforms of TERT protein originating from the alternate splicing events could be functionally important, as suggested for the *A. thaliana* variant TERT V(I8) (*TERT* variant in intron 8). This isoform of Arabidopsis TERT is able to bind Protection of telomeres protein 1a (POT1a), (one of the Arabidopsis orthologues of the human or fission yeast single-stranded telomeric sequence binding protein POT1) more efficiently than full-length TERT [156,157].

It has been proposed that human telomerase is subjected to posttranslational regulation, such as phosphorylation or ubiquitination [126,127], as reviewed in [112]. Putative phosphorylation sites were identified in TERT amino acid sequences from *O. sativa* [128] or *N. tabacum* [129], but not in TERT from *A. thaliana* [128].

In plants, the indirect regulation of telomerase by various proteins or hormones has also been described. In tobacco cell culture, phytohormones, such as abscisic acid or auxin, regulate the phosphorylation of telomerase protein, which is required for the generation of a functional telomerase complex [129,158]. In *A. thaliana*, reduced endogenous concentrations of auxin in telomerase activator 1 (TAC1) mutant plants block the ability of this zinc-finger protein to induce TERT. However, Arabidopsis TAC1 does not directly bind the *TERT* promoter [159,160]. Similar to humans, Arabidopsis RuvBL2a protein may be involved in the regulation of *TERT* transcription in plants because in RuvBL2a heterozygous mutants, a moderate but significant increase in *TERT* transcripts was observed. Interestingly, telomerase activity in these plants was reduced to ca. 5% compared to WT plants [134].

The regulation of telomerase activity may also be driven by the modulation of TR maturation. As described previously, biogenesis of the human TR involves a complex series of posttranscriptional modifications (Figure 4b, Table 1b) (reviewed in [84]). In humans, the set of *TR* transcripts with heterogenous 3′-ends may be trimmed by various exonucleases [161]. Similarly, various 5′ cap-binding complexes can be recruited to a mono- or tri-methylguanosine cap [82,162].

There is also evidence that in addition to its canonical role in telomere maintenance, both telomerase subunits—TERT and TR—can function independently of telomerase [163]. It was demonstrated that, e.g., the TR subunit was upregulated at very early stages of tumorigenesis, whereas telomerase activity was detected in end-stage tumors [164], and that the RNA component seems to be capable of DNA damage response (DDR) regulation [165]. The TERT subunit, independently of its action on telomeres, regulates the cell-cycle, inhibits apoptosis, regulates gene expression, modulates cell signaling (e.g., Wnt/β-catenin, NF-κB pathways) and DDR, and binds to and protects mitochondrial DNA (mtDNA) (reviewed in [85,163,166]).

In plants, the armadillo/β-catenin-like repeat-containing protein (ARM) or Chromatin remodeling 19 (CHR19) proteins associated with TERT may reflect the possible non-telomeric functions of telomerase [167]. ARM proteins play a role in the Wnt/β-catenin signaling pathway in humans [168], but the non-telomeric functions of plant TERT or TR remain elusive.

## 6. Composition of Enzymatically Active Telomerase

The active human telomerase is composed not only of a core complex of TR encircled by TERT, but is assembled as a functional complex in a stepwise regulated process governed by multiple stably- or transiently-associated proteins.

Human telomerase RNPs, as well as other box snoRNPs or scaRNPs, are associated with the conserved H/ACA boxes binding protein complex (dyskerin, NOP10, NHP2 and NAF1) in the nucleoplasm, where NAF1 is replaced by GAR1 before the TR RNP complex reaches the nucleolus (Figure 4c) (reviewed in [84,169]). The assembly of TR and TERT into catalytically active telomerase is aided by RuvBL1 (pontin) and RuvBL2 (reptin) AAA+ ATPases, due to their direct interaction with TERT and dyskerin [140]. In mammals, the telomerase RNP is retained in nucleoli through the interaction between TERT and nucleolin in the dense fibrillar component [137,170]. Telomerase activity is negatively regulated by the nucleolar protein PIN2/TERF1-interacting telomerase inhibitor 1 (PINX1) [171]. Nucleophosmin (NPM) and microspherule protein 2 (MCRS2) may be S phase-specific co-effectors of PINX1, working against each other to modulate the human TERT pool (reviewed in [172]). Telomerase is then recruited to Cajal bodies (CBs). CBs are spherical sub-nuclear organelles that reside at the nucleolar periphery and are implicated in RNA-related metabolic processes. TCAB1 (also known as WDR79, WRAP53), bound to the CAB-box motif of TR, promotes the translocation to CBs [96]. CB-related proteins, such as coilin and survival motor neuron (SMN), interact with telomerase and may regulate the formation of an active telomerase complex [141,173,174,175]. The CBs colocalize with telomeres and facilitate the recruitment of the mature telomerase complex to the telomeres via the telomere-associated protein TPP1, a subunit of the Shelterin complex localized at telomeres (Figure 4d) [144,176].

The broad conservation of a dyskerin–TR association was proposed among diverse organisms, including plants. For example, telomerase activity and TR were immunoprecipitated with the anti-dyskerin antibody in onions (*Allium cepa*) (Table 1c) [19]. In *A. thaliana*, null mutants for the nucleolar protein NUCLEOLIN 1 caused telomere shortening on all chromosomal arms, although a direct interaction between NUCLEOLIN 1 and TERT in Arabidopsis was not observed [138]. Similarly, we demonstrated that the plant RuvBL1 and RuvBL2a proteins interacted with TERT only indirectly in the nucleolus in vivo. In contrast to mammals, interactions between the TERT and RuvBL proteins in *A. thaliana* were not direct, but rather they were mediated by one of the Telomere Repeat Binding (TRB) proteins [134,177]. It was also shown that in *A. thaliana,* dyskerin directly interacted with NAF1. Plant dyskerin was localized not only in the nucleoli, but was also detected in CBs [136]. The main abundant signature protein of CBs in plants, as well as in mammals, is coilin (reviewed in [178]). Dvořáčková detected the significant colocalization of TRB1 with coilin present in the CBs adjacent to the nucleolus (Dvořáčková’s thesis). TRB proteins, which are the only proteins with confirmed in vivo plant telomere localization and function [145,179,180,181], may help to recruit telomerase to telomeres as they directly interact with TERT (Table 1d) [145]. Moreover, TRB proteins associate with the dyskerin orthologue CBF5 in the nucleolus, and they directly interact with POT1b (one of the plant paralogues of the Shelterin POT1 subunit). For a recent list of proteins associated with human and plant telomerase or with telomeric sequences, see Procházková Schrumpfová 2019 [7].

Thus, while telomerase-interacting proteins (reviewed in [172]) show relatively extensive conservation, individual interactions remain to be elucidated and carefully classified into direct and indirect ones. Due to the recently described differences in TR biogenesis pathways between plants and ciliates on one hand, and mammals and yeasts on the other hand, the orthologs of known TR interactors in humans (dyskerin, NOP10, NHP2, NAF1 or GAR1), as well as, e.g., orthologues of the La-family protein (p65) from ciliates, or Sm proteins from yeasts (reviewed in [64]), should also be examined in plants. Ideally, a new independent screen and subsequent analyses should identify plant TR direct interactors de novo.

## 7. Conclusions 

Here we have provided an updated overview of telomerase—its origin, biogenesis, regulation and function. Despite the extensive conservation of telomerase as a tool to overcome the end-replication problem of linear eukaryotic chromosomes, the subsequent evolution of this ancient molecular machine resulted in alternative solutions for particular aspects of its biogenesis and composition, as exemplified by the recent description of telomerase RNAs or telomerase-interacting proteins in land plants. The further investigation of telomerase diversity across the width of eukaryotic phylogeny is needed for a deeper understanding of truly fundamental principles of telomere and telomerase regulation, and the potential application of this knowledge in medicine, plant breeding or the protection of biodiversity.

## Abreviations

hhumanAAA+ ATPases associated with diverse cellular activitiesALTalternative mechanisms of lengthening of telomeres ARMarmadillo/β-catenin-like repeat-containing protein At
*Arabidopsis thaliana*
CAB-boxCajal body-box CBF5centromere-binding factor CR4/5conserved region 4/5 CRM1chromosome region maintenance 1 protein homolog CRuMscollodictyonids, Rigifilida, *Mantamonas*CTEC-terminal extension DDR DNA damage response DNA Pol αDNA polymerase alphaDNA Pol δ DNA polymerase deltaDNA Pol ε DNA polymerase epsiloneToLTree of Life FHCfetal human colon FLfull-length GAR1,2Glycine Arginine Rich 1, 2 H/ACA H/ACA (H-box (consensus ANANNA) and ACA-box (ACA))HRhomologous recombination Hsp70heat shock protein 70 Hsp90heat shock protein 90 HT-29adenocarcinoma colon CHIPcarboxyl-terminus of Hsp70 Interacting Protein CHR19chromatin remodeling 19 Impimportin LECAlast eukaryote common ancestor LUCAlast universal common ancestor MCRS2microspherule protein 2 MKRN1E3 ubiquitin-protein ligase makorin-1 MMGmonomethylguanosine mRNAmessenger RNAmtDNAmitochondrial DNA n.p.nuclear pores NAF1nuclear assembly factor 1 NCLnucleolin NF- κBnuclear factor κB NF-κBnuclear factor kappa-light-chain-enhancer of activated B cellsNHP2non-histone protein 2 NLSnucleus localization-like signal non-LTR-retrotransposons non-long-terminal-repeat retrotransposons NOP10nucleolar protein 10 NPMnucleophosmin NUC-L1 nucleolin-like1Pphosphorylation p23co-chaperone PINX1PIN2/TERF1—interacting telomerase inhibitor 1 poly (A) tail polyadenylate tail POT1protection of telomeres protein 1 POT1aprotection of telomeres protein 1a RAP1repressor/activator site binding protein RID1RNA interaction domain 1 RNA Pol IIRNA polymerase IIRNA Pol IIIRNA polymerase IIIRNPsribonucleoproteinsRTreverse transcriptase RT domainreverse transcriptase motifs domain RT-qPCR reverse transcription-quantitative PCRRuvBL1RuvB-like 1 AAA+ ATPases (pontin) RuvBL2RuvB-like 2 AAA+ ATPases (reptin) scaRNA small Cajal body RNA SHQ1snRNA of the box H/ACA family quantitative accumulation 1 SMNsurvival motor neuron protein snoRNA small nucleolar RNA SP1/3specificity protein 1/3 STAT3signal transducer and activator of Transcription 3 t/PKtemplate/pseudoknot TAC1telomerase activator 1 TBEtemplate boundary element TCAB1telomere cajal body protein 1 TENtelomerase essential N-terminal domain TERTcatalytic telomerase reverse transcriptase TIN2TRF1-interacting nuclear factor 2 TMGN2, 2, 7 trimethylguanosine TPP1TIN2- and POT1-organizing protein TR, TER, TERCtelomerase RNA component TRBDRNA-binding domain TRF1/2telomeric-repeat binding factor 1/2 TSARtelonemids, stramenopiles, alveolates, and RhizariaUbqubiquitin USEupstream sequence element Wnt/β-catenin wnt/beta-catenin- α TERTminus alpha TERT 

## Figures and Tables

**Figure 1 biomolecules-10-01425-f001:**
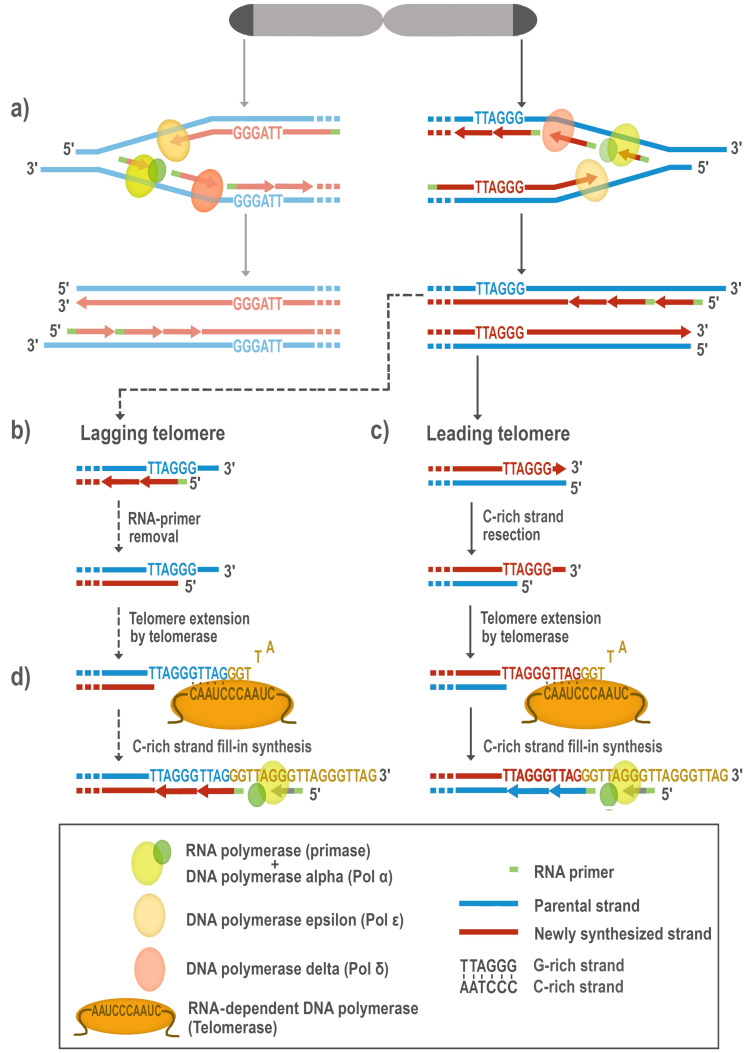
The replicating DNA in eukaryotes: DNA polymerases involved in replication. (**a**) During semiconservative DNA replication, each strand serves as a template for DNA polymerases to synthesize a new complementary strand. A specialized RNA polymerase (primase), that is a part of DNA Pol α, synthesizes the RNA primer. A single RNA primer aids DNA replication on the leading strand and multiple primers initiate Okazaki fragment synthesis on the lagging strand. Further DNA synthesis is carried out by DNA Pol ε and DNA Pol δ (reviewed in [4]). (**b**) The newly replicated telomere resulting from the lagging strand synthesis (Lagging telomere) retains the terminal RNA primer, which is subsequently removed. Attachment of the last RNA primer more proximally on the DNA strand, together with RNA-primer removal, creates an overhang on the G-rich strand. (**c**) The initial product of the leading strand DNA synthesis (Leading telomere) is a blunt terminus whose C-rich strand is then resected by an exonuclease to create the mature G-rich overhang. (**d**) In cells with an active RNA-dependent DNA polymerase (Telomerase), the G-rich overhangs, originating from Lagging or Leading telomeres, can undergo elongation (reviewed in [5]). Telomerase carries its own RNA molecule, which is used as a template, and can anneal through the first few nucleotides of its template region to the distal-most nucleotides of the G-rich overhang of the telomere DNA, add a new telomere repeat (GGTTAG) sequence, translocate, and then repeat the process. The complementary C-rich strand is then in-filled by DNA Pol α-primase [6].

**Figure 2 biomolecules-10-01425-f002:**
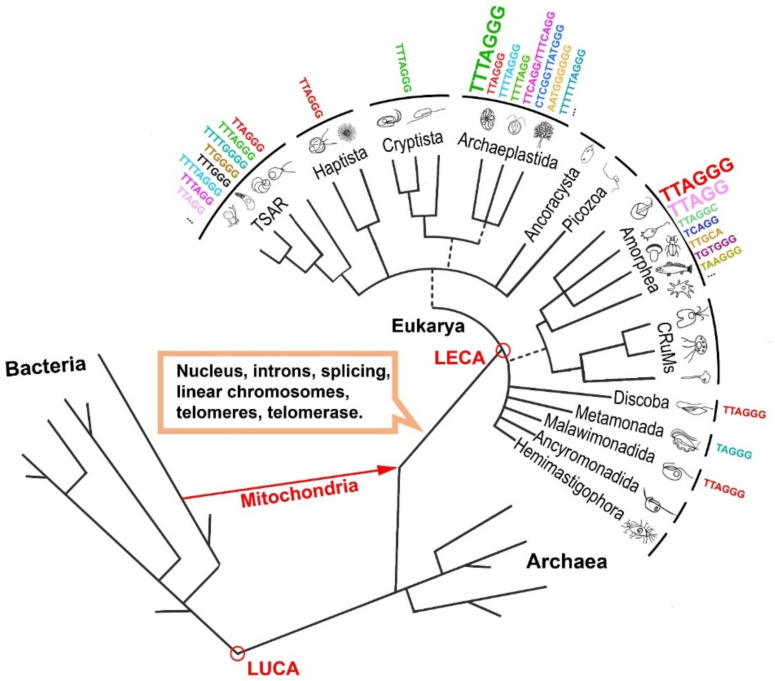
Telomeres and telomerase in the evolutionary tree. A simplified phylogenetic tree is shown, where telomeres and telomerase evolved upon linearization of chromosomes by the insertion of Group II self-splicing introns [37]. In the Eukaryote branch, the groupings correspond to the current ‘supergroups’ according to the recent eukaryotic Tree of Life (eToL) [24]. Unresolved branching orders among lineages are shown as multifurcations. Broken lines reflect lesser uncertainties about the monophyly of certain groups. Examples of known telomeric repeat variants are listed next to respective supergroups (see also Appendix A). The major known telomeric repeat variants in the supergroups are marked with a larger font [22,36,38] (see text for details). Last eukaryote common ancestor (LECA); last universal common ancestor (LUCA); telonemids, stramenopiles, alveolates and Rhizaria (TSAR); collodictyonids, Rigifilida, *Mantamonas* (CRuMs). The living species icons are partly adopted from Adl et al., 2012 [39].

**Figure 3 biomolecules-10-01425-f003:**
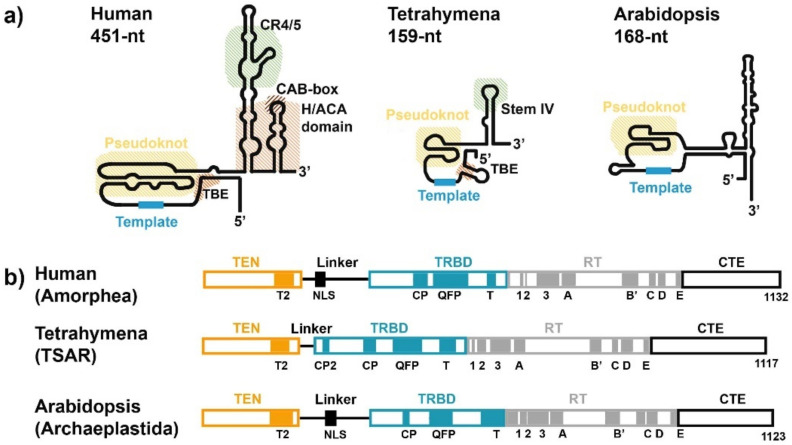
Conservation of functional domains of two core telomerase subunits—TERT and TR. (**a**) Models of secondary structures of human, Tetrahymena and Arabidopsis TRs suggest the conservation of several structural motives, including pseudoknot in the vicinity of the template (t/PK domain) and stem-loop regions [70,71]. In humans the stem-loop region contains the conserved 4/5 (CR4/5) region, the H (AnAnnA) and ACA-boxes (H/ACA) domains and the Cajal body box (CAB-box) motif that serve as binding sites for other protein components of the telomerase holoenzyme complex (dyskerin, NOP10, NHP2 and GAR1). In Tetrahymena, the stem-loop 4 (SLIV) is directly bound by p65 protein [72]. To date, particular interactors and their binding sites have not been demonstrated directly in Arabidopsis (see also Table 1). (**b**) Domain arrangement of human (animals), Tetrahymena (ciliates) and Arabidopsis (plants) TERTs. The supergroup for each species is given (see Figure 2). N-terminus: the telomerase essential N-terminal (TEN) domain and RNA-binding domain (TRBD domain) are separated by a Linker that contains a nucleus localization-like signal (NLS). The central RT domain: catalytical part of the enzyme that contains seven evolutionary-conserved RT motifs (1, 2, A, B′, C, D and E motifs) and also telomerase specific 3 motif [73,74,75]. C-terminus: C-terminal extension (CTE) domain.

**Figure 4 biomolecules-10-01425-f004:**
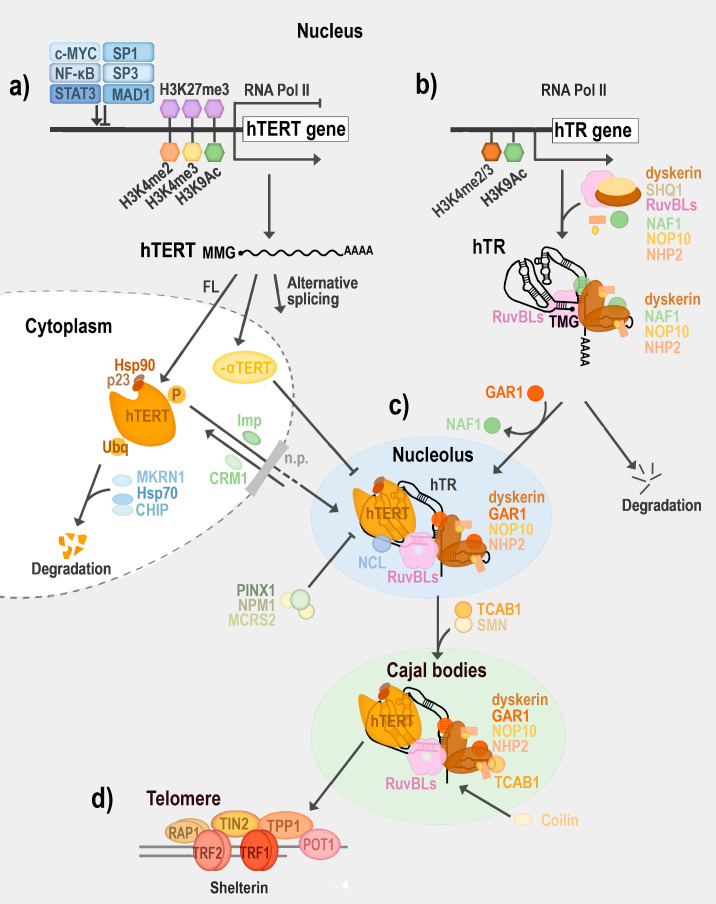
Regulation of human telomerase biogenesis. (**a**) Transcription of the human telomerase reverse transcriptase gene (*hTERT*) by RNA polymerase II (RNA Pol II) is regulated by several activators and repressors acting at the promoter level (e.g., c-MYC, Nuclear Factor κB (NF- κB), Signal Transducer and Activator of Transcription 3 (STAT3), Specificity Protein 1/3 (SP1/3), MAD1). Histone modification H3K27me3 often silences *hTERT*, however the mutated *hTERT* allele is marked by the active histone marks H3K4me2, H3K4me3 and H3K9ac. *hTERT* pre-mRNA with a 5′ mono-methylguanosine (MMG) cap and poly(A) 3’ tail can be spliced into full-length (FL) or multiple alternative isoforms (Alternative splicing) that are catalytically inactive or even inhibit telomerase activity (e.g., minus alpha hTERT (-α hTERT) due to their competition for human TR (hTR) with FL hTERT). The binding of heat shock protein 90 (Hsp90) with its co-chaperone (p23) in the cytoplasm enables hTERT phosphorylation (P). hTERT is further imported back to the nucleus by Importin α or β1 (Imp) via nuclear pores (n.p.), while the export of hTERT may be mediated by the chromosome region maintenance 1 protein homolog (CRM1, also known as exportin-1). The ubiquitin (Ubq)-proteasomal degradation of TERT is driven by E3 ubiquitin-protein ligase makorin-1 (MKRN1), heat shock protein 70 (Hsp70) and carboxyl-terminus of Hsp70 Interacting Protein (CHIP). (**b**) Histone modifications H3K4me2/3 or H3K9Ac help to regulate read-through of the human telomerase RNA (hTR) gene by RNA Pol II in telomerase-positive cell lines. SHQ1 chaperone and RuvB-like proteins (RuvBLs) facilitate the assembly of nascent RNA with RNA scaffold proteins (dyskerin, NOP10, NHP2, and NAF1). Mature hTR is capped with a tri-methylguanosine (TMG) cap at the 5′ end, polyadenylated at the 3′ end and co-transcriptionally associated with scaffold proteins. The hTR variants with shorter or longer 3′ ends, or those associated with variant proteins, may lead to the degradation of hTR. NAF1 is replaced by GAR1 before the hTR ribonucleoprotein complex reaches the nucleolus. (**c**) RuvBLs (pontin and reptin) enable telomerase assembly and allow hTERT recruitment to the nucleolus to form a mature telomerase complex while bound by nucleolin (NCL). PIN2/TERF1-interacting telomerase inhibitor 1 (PINX1), together with nucleophosmin (NPM) and microspherule protein 2 (MCRS2), regulate hTERT availability in a cell cycle-dependent manner. Telomere Cajal body protein 1 (TCAB1, also known as WRAP53) recognizes the Cajal body box (CAB-box) of the hTR in the mature telomerase complex and recruits it to the Cajal bodies (CBs). In CBs, hTR interacts with local proteins such as coilin while survival motor neuron protein (SMN) binds hTERT. (**d**) In the S-phase, the CBs colocalize with telomeres and facilitate the recruitment of the mature telomerase complex to the telomeres via interaction with TPP1 protein, which is one of the subunits of a protein complex localized at telomeres, termed as Shelterin. The presence of Shelterin proteins (telomeric-repeat binding factor 1/2 (TRF1/2), protection of telomeres protein 1 (POT1), TRF1-interacting nuclear factor 2 (TIN2), repressor/activator site binding protein (RAP1) and TPP1) helps distinguish chromosomal ends (telomeres) from DNA breaks. (For references see Text or Table 1).

**Table 1 biomolecules-10-01425-t001:** Human and Arabidopsis telomerase assembly—a comparative overview (a–d classification corresponds to Figure 4).

		Mammals (Human)	Reference(s)	Plants (*Arabidopsis thaliana*)	Reference(s)
**(a) TERT**	**Minimal promoter**	330 bp upstream of the translation start site to 228 bp downstream.	[115,116,121]	336 bp long promoter region of the translation start site with plausible regulatory intron 1.	[13,117]
	**RNA Polymerase**	RNA Pol II	[115]	RNA Pol II	[122]
	**Histone modifications of promoter**	Telomerase-negative tissues: H3K27me3; telomerase-positive tissues (mutated *TERT* allele): H3K4me2, H3K4me3 and H3K9ac.	[118,119,123]	Telomerase-negative tissues: H3K27me3, H3K4me3, H3K9Ac; telomerase-positive tissues: H3K4me3, H3K9Ac.	[11]
	**TERT expression in organism**	TERT expression is strictly controlled at the transcript level.	[15,124]	The dynamics of *TERT* transcripts correlates with telomerase activity observed in plant tissues.	[7,11]
	**Number of exons**	16 exons	[75,121]	12 exons	[75]
	**Alternative splicing of mRNA**	*TERT* pre-mRNA can be spliced into at least 22 isoforms.	[125]	*TERT* pre-mRNA can be spliced into 3 isoforms.	[75]
	**Post-translational modifications**	Phosphorylation or ubiquitination.	[126,127]	No putative phosphorylation site in *A. thaliana* TERT (but predicted in rice or tabacum TERT ).	[128,129]
	**Import to the cell nucleus**	Importin α promotes nuclear import of the TERT.	[130]	Importin subunit alpha-4 is associated with TERT.	[98]
	**Protein domains**	TEN, TRBD, RT, CTE.	[75,108]	TEN, TRBD, RT, CTE.	[75]
	**Protein length**	1132 aa	[108]	1123 aa	[131]
**(b) TR**	**Histone modifications**	TR expression in telomerase-positive cell lines is associated with H3K4me2/3, H3K9Ac and hyperacetylation of H4.	[132,133]	Not known yet.	
	**RNA Polymerase**	RNA Pol II	[66]	RNA Pol III	[19,20]
	**Modifications**	5′ end cap, internally modified, poly (A) tail	[83]	Not known yet.	
	**Template region**	11 nt long template region (synthesizes 6 nt telomeric repeats GGTTAG).	[66,88]	9 nt long template region (synthesizes 7 nt telomeric repeat GGTTAG).	[19]
	**TR gene length**	451 nt long transcript	[66]	268 nt long transcript	[19,20,71]
	**TR expression in organism**	In most tissues TR is ubiquitously expressed regardless of telomerase activity.	[16,17]	The dynamics of *TR* transcripts correlates with telomerase activity observed in plant tissues.	[7,11]
**(c) Nucleolus and CBs**	**TR scaffold proteins**	Dyskerin, NOP10, NHP2, NAF1/GAR1.	[84,96]	Not known yet. Dyskerin (CBF5), NOP10, NHP2, NAF1, and GAR1 are localized in the nucleolus. Telomerase activity can be immunoprecipitated with dyskerin (CBF5) in plants. Dyskerin associates with TRB proteins.	[19,134,135,136]
	**Nucleolin**	NCL involves nucleolar localization of TERT.	[137]	NUC-L1 has a role in telomere maintenance and telomere clustering.	[138,139]
	**RuvBLs**	RuvBLs (pontin and reptin) interact with TERT and dyskerin.	[140]	Interactions between TERT and RuvBL proteins are mediated by TRB proteins.	[134]
	**coilin**	Interacts with TR.	[141,142]	Colocalizes with TRB1 in the CBs adjacent to the nucleolus.	[143]
**(d) Association with telomere**		The TPP1 protein interacts with TERT and facilitates the recruitment of the mature telomerase complex to the telomeres.	[144]	The TRB proteins interact with TERT and may help to recruit telomerase to the plant telomeres.	[145]

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
