# Peer review of "Composition and Function of Telomerase—A Polymerase Associated with the Origin of Eukaryotes"

_biomolecules, 2020, doi:10.3390/biom10101425_

Round 1
Reviewer 1 Report
Comments to the authors:
The authors review the composition and function of telomerase across eukaryotes. The presented results or explanations on telomerase are of great interest and importance to the field and it is very interesting to see how differences in telomerase composition and activity between eukaryotic groups may have evolved. The review is very well written, well-structured and very detailed and I liked that alternative functions of telomerase are also part of this review. It was a pleasure to read this manuscript and I only have a few minor comments, and suggestions to improve this review (see below).
Comment 1: A list of abbreviations or a glossary would make the manuscript easier to read and understand
Figure 1: Is it possible to devide the figure in a, b and c. I think this would make it easier to understand
Line 44: A single RNA primer … This is not shown in the figure –right? Could you maybe indicate this
Line 49: G-rich strand -> in the figure legend it is just called G-stand
Line 101: telomeric runs -> maybe patterns or motives would be a better word.
Figure 2: please specify TSAR and CRuMs in the figure text
Line 148: please give ribonucleoprotein for RNPs once in the text. I think a list of abbreviations would help.
Author Response
Response to Reviewer 1 Comments
Point 1: Comment 1: A list of abbreviations or a glossary would make the manuscript easier to read and understand
Response 1: The list of abbreviations was added.
Point 2: Figure 1: Is it possible to devide the figure in a, b and c. I think this would make it easier to understand
Response 2: The Figure 1 was divided into parts a), b), c) and d).
Point 3: Line 44: A single RNA primer … This is not shown in the figure –right? Could you maybe indicate this.
Response 3: A single RNA primer that adds DNA replication on the leading strand is indicated in the Figure 1.
Point 3: Line 49: G-rich strand -> in the figure legend it is just called G-stand
Response 4: The abbreviation G-strand and C-strand was replaced by G-rich strand and C-rich strand.
Point 4: Line 101: telomeric runs -> maybe patterns or motives would be a better word.
Response 5: The word runs is replaced by word motifs.
Point 5: Figure 2: please specify TSAR and CRuMs in the figure text
Response 5: The abbreviations were added to the text.
Point 6: Line 148: please give ribonucleoprotein for RNPs once in the text. I think a list of abbreviations would help.
Response 6: The abbreviation RNP was added to the text. The list of abbreviations was added to the manuscript.
Reviewer 2 Report
Thank you for the opportunity to review the manuscript entitled “Composition and Function of Telomerase – a 3 polymerase associated with the origin of eukaryotes” by Schrumpfová & Fajkus.
The manuscript reviews the literature with latest details on function of telomerases in its glorious diversity; including a discussion on origin and phylogenetic divergence. It contains very useful comparative overviews, and is generally a very useful source of material. It’s timely, including the latest developments in plants.
A couple of comments:
Point 4 (TERT Subunit of Telomerase) appears mostly as a list of facts. A bit more integration/opinion/discussion would benefit the quality of the review.
First paragraph page 6. The author states: “Previously characterized TER1 and TER2 in A. thaliana were shown not to act as telomerase RNAs [19,67] and the original paper describing them has been retracted [68,69]”. Why was it brought up? –describe in the text a bit more.
Author Response
Response to Reviewer 2 Comments
Point 1: (TERT Subunit of Telomerase) appears mostly as a list of facts. A bit more integration/opinion/discussion would benefit the quality of the review.
Response 1: In this review, we intentionally wrote a separate chapter The Origin of Telomerase which precedes the description of known facts on TR and TERT telomerase subunits, and presents hypotheses on the origin of telomerase. We thus clearly separated hypotheses from the accumulated knowledge. In a couple of years, hopefully, we will be able to present a more integrated view on plant telomerase – but the current state of knowledge does not allow us to do so.
Point 2: First paragraph page 6. The author states: “Previously characterized TER1 and TER2 in A. thaliana were shown not to act as telomerase RNAs [19,67] and the original paper describing them has been retracted [68,69]”. Why was it brought up? –describe in the text a bit more.
Response 2: We prefer not to expand this part with more details as the corresponding author of the original (now retracted) paper – Dorothy Shippen – could possibly consider it offensive.